# A PET Fluorescent Probe for Dynamic Pd^2+^ Tracking with Imaging Applications in the Nanofiber and Living Cells

**DOI:** 10.3390/molecules28073065

**Published:** 2023-03-29

**Authors:** Zhao Cheng, Xilang Jin, Yinggang Liu, Xuejiao Zhang

**Affiliations:** 1School of Pharmacy, Xi’an Medical University, Xi’an 710021, China; 2School of Materials and Chemical Engineering, Xi’an Technological University, Xi’an 710032, China

**Keywords:** palladium ion, imaging, nanofiber, cytotoxicity, mechanism

## Abstract

Constructed on the moiety of a lactam screw ring, a near-infrared fluorescent probe RCya for Pd^2+^ was designed under the PET mechanism and synthesized by incorporating 2,4-dihydroxybenzaldehyde as the recognition group. Dynamic detection of aqueous Pd^2+^ by the probe RCya could be accomplished through ion competition, linear response, fluorescence-pH/time stabilities, and other optical tests. Moreover, the high selectivity, low cytotoxicity, cell permeability, and lysosome accumulation properties of RCya enabled the imaging applications on solid-state RCya–PAN composite nanofibers and in living cells. The recognition mechanism of probe RCya toward Pd^2+^ was further studied through simulation calculation and MS analysis.

## 1. Introduction

Palladium, as a mature and early-used platinum-group catalyst, has a wide distribution in the soil, water, and other environmental sources. When entering the food chain through contaminated food or water and being taken into the human bodies, the food-borne intake palladium would not only accumulate in the organs but bound with thio-containing amino acids, proteins, DNA, and other related biomolecules [1], disrupting a variety of biological processes as a biological participant. For its toxicity, allergenicity, carcinogenicity, and therapeutic and biological effects [2,3], effective and quantitative methods for low-level and intracellular palladium detections are urgently needed. Moreover, ppm analytical accuracy should be achieved in the palladium detections, considering the proposed dietary palladium intake of 1.5–15 μg per person per day and the 5–10 ppm governmental regulations on palladium in end products/drugs [4].

Compared with the instrumental analyses with complex requirements on instruments and sample preparations, optical detection approaches, especially the colorimetric and fluorimetric analytical approaches [5,6], demonstrated unique advantages in low cost, simple operation, high sensitivity, and so on. However, most optical detections were reported to happen in the aqueous environment and required a good solubility of the related fluorescence probe. For more convenience and more future applications in target detection, more visualized detection methods between the probe and the target, such as naked-eye mode [7], solid-state detection, and dynamic imaging [8], should be adopted in the structural design of fluorescent probes.

Accordingly, a near-infrared fluorescent probe RCya for Pd^2+^ was synthesized by introducing the recognition group 2,4-dihydroxybenzaldehyde into the moiety of a lactam screw ring [9]. In an aqueous environment, probe RCya demonstrated specific responses toward Pd^2+^ over other metals, accompanied by a good quantitative relationship, good fluorescence-pH/time stabilities, and remarkable changes in UV-Vis/fluorescence spectra of RCya–Pd^2+^. Besides the dynamic Pd^2+^ detection in aqueous solutions, colorimetric and specific detection of Pd^2+^ by RCya could also be accomplished on solid-state RCya–PAN composite nanofibers. Furthermore, MTT and bioimaging in living cells indicated the low cytotoxicity, good cell permeability, and lysosome accumulation properties of probe RCya, enabling the dynamic labeling and tracking of Pd^2+^ for instant and time-lapse imaging in biological applications, also predicting the probe’s future applications as real-time detection methods for Pd^2+^ in clinical diagnosis.

## 2. Results

### 2.1. Structural Characterization

With a four-step synthetic route, the fluorescent probe RCya for Pd^2+^ was synthesized. Structures of RCya and related intermediates were characterized by ^1^H NMR and HRMS. The calculation and measurement results of MS were all matched, and characteristic ^1^H NMR chemical shifts were all located at the reasonable ppm range and in accordance with the target structures.

### 2.2. Spectroscopic Properties

#### 2.2.1. Selectivity and Competition

A specific detection for biological Pd^2+^, for palladium ions’ complex ions coexistence environment, is a difficult issue, and the parameters of selectivity/competition must be taken into consideration.

For the detection of aqueous Pd^2+^, the selectivity of probe RCya toward Pd^2+^ was evaluated in PBS buffer solution (EtOH : H_2_O- = 1 : 99, pH = 7.4) with the concentration of RCya 10 μmol L^−1^ and metal ions 40 μmol L^−1^. When excited by 635 nm, probe RCya itself displayed maximal fluorescence intensity at 735 nm and maximal UV-Vis absorbance at 713 nm, respectively. Upon the addition of various metal ions, as illustrated in the front rows of Figure 1a,b, there were obvious differences in these RCya–metal ion systems, among which only Pd^2+^ immediately triggered a significant optical enhancement at 735 nm for fluorescence and 713 nm for UV-Vis, with nearly no interferences from other metal ions.

Subsequently, 40 μmol L^−1^ Pd^2+^ was added to the front-row-represented RCya–metal ion solutions, resulting in ions coexistence systems of RCya–metal ion-Pd^2+^. From the systems of RCya–metal ion to RCya–metal ion-Pd^2+^, the resulting enhancements in 735 nm (fluorescence) and 713 nm (UV-Vis) peaks were depicted in the rear rows of Figure 1a,b. Compared with the front rows, it was undoubtedly the subsequently added Pd^2+^ that induced significant increases in the RCya–metal ion-Pd^2+^ systems. Meanwhile, for every RCya–metal ion-Pd^2+^ system, optical intensities of 735 nm (fluorescence) and 713 nm (UV-Vis) were enhanced of almost the same fold, which meant every system’s enhancement was caused by the addition of Pd^2+,^ and the existence of other metal ions had nearly no competitions or influences on the specific optical responses of RCya–Pd^2+^.

The selectivity and competition tests showed that probe RCya was highly sensitive to Pd^2+^, predicting the future use of RCya for Pd^2+^ detection in complex environments, which will not be affected by the tested background of other metal ions.

#### 2.2.2. Linear Relationship

For a better understanding of the quantitative relationship between probe RCya and Pd^2+^, fluorescence and UV-Vis titration experiments were performed with 10 μmol L^−1^ RCya and 0.0–4.0 equiv. Pd^2+^ to RCya. As 0.0–4.0 equiv. Pd^2+^ was gradually added into RCya, a significant enhancement of RCya–Pd^2+^ located at 735 nm for fluorescence (Figure 2a) and 713 nm for UV-Vis absorbance (Figure 2b) resulted.

Analysis of the fluorescence spectra of RCya–Pd^2+^ resulted in a linear relationship of the fluorescence intensity of RCya versus [Pd^2+^] in the concentration range of 0–40 μmol L^−1^ Pd^2+^ [10], with the linear equation seen in Figure 3a as y = 2.00737 × 10^7^ + 5.58777 × 10^6^ x (R^2^ = 0.99622) (y, I_735_: fluorescence intensity of the RCya–Pd^2+^ system at the wavelength of 735 nm; x, [Pd^2+^]: the concentration of Pd^2+^), indicating a highly quantitative detection of Pd^2+^ by probe RCya. Furthermore, based on the fluorescence titration results and the formula for the limit of detection LOD = 3σ/k, the detection limit of Pd^2+^ by RCya could be determined. In the detection limit equation, the fluorescence emission spectra of RCya in the absence and presence of Pd^2+^ were measured three times, and the standard deviation of blank fluorescence measurement (σ in the detection limit equation) was obtained, whereas the slop k in the equation was calculated through the plot of fluorescence intensity at 735 nm versus Pd^2+^ concentration. The detection limit was calculated to be 0.362 μmol L^−1^, indicating a high sensitivity of RCya toward Pd^2+^.

Similarly, the linear response of RCya’s UV-Vis absorbance toward [Pd^2+^] (0–40 μmol L^−1^) could be expressed by the following equation [11]: y = 0.00292 x − 0.0122 (R^2^ = 0.9943) (y, Abs_713_: UV-Vis absorbance of the RCya–Pd^2+^ system at the wavelength of 713 nm; x, [Pd^2+^]: the concentration of Pd^2+^), as depicted in Figure 3b.

#### 2.2.3. Influence Parameters

For dynamic Pd^2+^ labeling and tracking in biological systems, fluorescence-time stabilities of the RCya–Pd^2+^ system should be analyzed to achieve dynamic and time-lapse detections. Initially, there underwent a recognition process of RCya and Pd^2+^ with a time length of about 40 s, (a) fluorescence intensity at 735 nm, and (b) UV-Vis absorbance at 713 nm, increasing gradually. After a response time of 0–40 s, optical intensities were then saturated and remained steady for minutes (Figure 4), indicating the probe RCya’s not only instant but time-lapse fluorescence imaging applications for dynamic Pd^2+^ detections in biological systems.

When probe RCya was applied to Pd^2+^ detection in biological systems, fluorescence-pH stabilities of RCya and RCya + Pd^2+^ should also be examined for the complex and variable pH environment. In comparison with probe RCya’s steady fluorescence intensity at 735 nm, as seen in Figure 5, at a varied pH region of 3.0 to 9.0, fluorescence intensity at 735 nm of RCya–Pd^2+^ initially kept relatively steady and gradually decreased (Figure 5), suggesting an appropriate pH range of 3.0–7.0 for Pd^2+^ detection by RCya, which will satisfy most physiological pH environment for Pd^2+^ detections [12]. The differences between RCya and RCya + Pd^2+^ in fluorescence-pH stabilities could be explained by the recognition mechanism of RCya and Pd^2+^. When probe RCya combined with Pd^2+^, the probe’s structure partly transformed, and the synthetic intermediate compound **3** was released. As a rhodamine derivative, compound **3** only gave off steady fluorescence signals at the acidic conditions, which was in accordance with the stabilities of RCya + Pd^2+^ at the pH range of 3.0–7.0. Once compound **3** was located in the basic environment, the rhodamine moiety became unstable, and its fluorescence dramatically decreased, consistent with the phenomenon of RCya + Pd^2+^ beyond pH = 7.

### 2.3. Mechanism

#### 2.3.1. Proposed Mechanism

Illustrated in Figure 1a and Figure 2a and partly revealed by the fluorescence-pH stabilities of RCya and RCya + Pd^2+^ in Figure 5, before and after the combination of Pd^2+^ by RCya, the probe’s structural transformation and the relevant changes in fluorescence signals, accompanied with a release of the rhodamine derivative compound **3**, could be inferred [13,14] (Figure 6).

Before the addition of Pd^2+^ into RCya, probe RCya, with a strong intramolecular PET process [15], only gave off a very weak fluorescence at 735 nm (Figure 1a); even when probe RCya was located at a varied pH region of 3.0 to 9.0 (Figure 5), the fluorescence emission kept steady due to its stable structure. When Pd^2+^ was added into RCya and the probe combined with Pd^2+^, the amide carbonyl group of the probe coordinated with Pd^2+,^ and, thus, RCya–Pd^2+^ combined, the lactam ring in RCya was opened due to the formation of complex RCya–Pd^2+^, which not only inhibited the PET process but emitted the specific rhodamine red fluorescence and the fluorescence emission intensity at 735 nm enhanced significantly (Figure 1a). Similarly, as 0.0–4.0 μmol L^−1^ Pd^2+^ was gradually added into RCya, the processes of PET-off and rhodamine ring-opened moiety would intensify, corresponding to the gradual fluorescence enhancements at 735 nm in Figure 2a.

Meanwhile, the combination process resulted in a ring-opened rhodamine moiety, which was highly similar to the synthetic intermediate compound **3** (Figure 1), as illustrated in the proposed recognition mechanism of Figure 6.

#### 2.3.2. DFT Calculation

For an extended evaluation of the photo-induced luminescence process [16,17], DFT calculations were employed and performed using the Gaussian 09 program. With the molecular models constructed on the corresponding relations of 6–31G* for elements C and H, 6–31 + G* for elements N, O, and S, and DGDZVP for Pd, the simulated diagram for the theoretical photo-induced luminescence process before and after Pd^2+^ combination by RCya was illustrated in Figure 7, which provided sufficient information on HOMO–LUMO gaps, combination sites, and the chemical activation process, further confirming the proposed mechanism.

In the free state of probe RCya, the HOMO and LUMO of RCya itself were located at the benzopyrrole fragment of the fluorophore and 2,4-dihydroxybenzaldehyde fragment of the recognition site, respectively. In comparison, the HOMO of the complex Rcya–Pd^2+^ was distributed entirely along the fluorophore, and the LUMO was still mainly located at the recognition site. The separation of HOMO and LUMO means possibilities for probe RCya to be excited. Furthermore, the calculated HOMO–LUMO energy gap for probe RCya was 2.81 eV, with a much smaller value of 1.67 eV for complex RCya–Pd^2+^, indicating that the complex RCya–Pd^2+^ located at a lower energy level, the combination process of RCya and Pd^2+^ was accompanied by a shift from a high energy level of the reactant RCya itself to a lower one of the product RCya–Pd^2+^, enabling the resulting complex RCya–Pd^2+^ with more stabilities. Additionally, due to its lower energy level and more stabilities, the complex RCya–Pd^2+^ was much more easily excited than probe RCya and gave off a much stronger fluorescence emission, in accordance with the optical properties depicted in Figure 1 and Figure 2.

#### 2.3.3. MS Analysis

Apart from the theoretical calculations, a further HRMS analysis offered us more details concerning the structural transformation from probe RCya to coordination complex RCya–Pd^2+^. When Pd^2+^ was added into RCya and the complex RCya + Pd^2+^ formed, a diagnostic peak at *m*/*z* = 559.2883 was obtained (Figure 8b), which was very close to *m*/*z* = 560.3299 [compound **3** + H^+^] of the synthetic intermediate compound **3** (Figure 8c). Through the combination process of RCya and Pd^2+^, the target peak *m*/*z* = 693.3237 [RCya+H^+^] of probe RCya (Figure 8a) was shifted to *m*/*z* = 559.2883 for RCya + Pd^2+^ (Figure 8b), indicating that the fragment of 2,4-dihydroxybenzene was eliminated when the lactam ring of RCya opened to capture Pd^2+^ and, meanwhile, the structure of precursor compound **3** was given back. Finally, the processes of lactam ring-opening, Pd^2+^ combination, and 2,4-dihydroxybenzene elimination combined to induce the tested *m*/*z* = 559.2883 in Figure 8b, corresponding to the proposed mechanism in Figure 6.

### 2.4. Nanofiber Imaging Applications

For the detection and imaging of Pd^2+^ by RCya on solid-state, RCya–PAN composite nanofibers were prepared and immersed into Pd^2+^ (5.0 μmol L^−1^)/other cations solution (100 μmol L^−1^), respectively, then imaged under ambient light or under the 365 nm ultraviolet lamp (Figure 9). Not surprisingly, only Pd^2+^ induced a colorless-to-red colorimetric and light yellow-to-green fluorescence change, accomplishing a simultaneous naked-eye Pd^2+^ detection under ambient light and spectroscopic Pd^2+^ detection under a UV lamp.

### 2.5. MTT Assay

A standard MTT assay was performed with HeLa cells to evaluate the potential imaging and diagnostic applications [18,19] of probe RCya. MTT assay was performed in HeLa cells with a concentration gradient of 6.25, 12.50, 25.00, 50.00, and 100.00 μmol L^−1^ for each group, respectively. The illustrations in Figure 10 indicated that there was no significant toxic effect of RCya on HeLa cells when incubated for 24 h. At the spectroscopic testing concentration of RCya (10 μmol L^−1^), HeLa cell viability was about 95% for RCya, even when the concentration of RCya reached the highest 100 μmol L^−1^ in MTT assay, the cytotoxicity of HeLa cells remained below 25%, indicating relatively low cytotoxicity of RCya at the concentration range of 0–100 μmol L^−1^, which satisfies the cytotoxicity concentration demands of ppm/μmol L^−1^ Pd^2+^ distributions in most biological systems.

### 2.6. Bioimaging

As seen in Figure 11a, an initial 30 min incubation of HeLa and 10 μmol L^−1^ RCya only gave off weak green fluorescence. When 40 μmol L^−1^ Pd^2+^ was subsequently added in and RCya + Pd^2+^ was incubated with HeLa for another 2 h at 37 °C, in comparison with Figure 11a, bright and specific red fluorescence of the rhodamine moiety was emitted from the intracellular area (Figure 11c), demonstrating the recognition of added Pd^2+^ by RCya over a very short time. The overlay of the bright-field image (Figure 11b) and fluorescence image (Figure 11c) are shown in Figure 11d, which located the red fluorescence signals in the perinuclear region of the cytosol [20,21], indicating that probe RCya was cell permeable and could be applied to the dynamic imaging of Pd^2+^ in living cells [22].

As implied by the bioimaging results in Figure 11, the accumulated areas of RCya in the cytoplasm were further analyzed in HeLa cells. By incorporating commercially available localization reagents Mito-Tracker Green (Mito) and Lyso-Tracker Green (Lyso), the co-staining experiments of RCya–Mito/Lyso were performed, and lysosome targeting properties of the synthesized RCya were revealed. As displayed in Figure 12a, RCya + Pd^2+^ and the lysosome-targeting reagent Lyso-Tracker Green demonstrated a good overlap, presenting large yellow patches. In contrast with Figure 12a, RCya + Pd^2+^ and the mitochondria-targeting reagent Mito-Tracker Green only gave a poor overlap, shown as the red/green patches in Figure 12b. Consequently, the co-staining imaging indicated that RCya could be used as an efficient lysosome-targetable probe for dynamic Pd^2+^ imaging in living cells, providing potential tools for monitoring the function of lysosomes in the autophagy processes.

## 3. Discussion

With the structure characterized by HRMS/^1^H NMR and optical properties analyzed by UV-Vis/fluorescence imaging, a fluorescent probe RCya was proven quite useful for the dynamic detection of Pd^2+^ in the lysosomes in real time. On the one side, highly specific responses to Pd^2+^ over other metal ions were accomplished by the probe, and the detections of Pd^2+^ by RCya not only happened in aqueous solutions but on solid-state nanofibers and in living cells. On the other, the combination mechanism and structural transformation of probe RCya in the recognition process of Pd^2+^ were studied through DFT calculations and HRMS analysis, offering us sufficient information on the structural design of fluorescent probes in future research. Furthermore, RCya–Pd^2+^ demonstrated good fluorescence stability at a relatively wide physiological pH range and in a considerably great time span; also, RCya–Pd^2+^ showed low cytotoxicity, good cell permeability, and lysosome-targeted properties in the MTT assay and dynamic bioimaging, which, combined, could predict probe RCya be appropriate for the biological applications under complex environments.

## 4. Materials and Methods

### 4.1. Reagents and Equipment

^1^H NMR spectra were recorded on a Varian INOVA-400 spectrometer at 400 MHz, and chemical shifts were reported relative to internal standard tetramethylsilane (TMS). MS analyses were carried out using Bruker micro TOF-Q II mass spectrophotometer. Fluorescent spectra were measured with a HITACHI F-4500 fluorescence spectrophotometer. UV-Vis spectra were measured on a SHIMADZU UV-2550 spectrometer. FT-IR spectra were recorded with KBr pellets on Bruker EQUINOX-55 FT-IR spectrometer. Results of cytotoxicity were analyzed with the Softmax pro software (version 2.2.1) in Spectra max190-Molecular devices. Cell-imaging experiments were performed using OLYMPUS FV1000 TY1318 laser scanning confocal microscope.

All the reagents and solvents used for synthesis were commercially available and used without further purification unless otherwise noted. Mito-Tracker Green, Lyso-Tracker Green, and HeLa cells were purchased from KeyGen BioTECH (Nanjing, China). The reaction process was monitored by thin-layer chromatography (TLC) on silica gel GF254. The products were purified by column chromatography on Merck silica gel (250–400 mesh ASTM). Phosphate Buffered Saline (PBS, pH = 7.4) was purchased from a Sinopharm chemical reagent company, Shanghai, China. Double-distilled water was used throughout the process of solution preparation and spectroscopic testing. Solutions of metal ions were prepared from their nitrate and chloride salts, PdCl_2_, SnCl_2_, CuCl_2_, MgCl_2_, Cr(NO_3_)_3_, CdCl_2_, CoCl_2_, BaCl_2_, PbCl_2_, AgNO_3_, FeCl_3_, FeCl_2_, NiCl_2_, AlCl_3_, ZnCl_2_, MnCl_2_, NaCl and KCl.

### 4.2. Synthesis

By introducing 2,4-dihydroxybenzaldehyde as the recognition group into the moiety of a lactam screw ring [23,24], a near-infrared fluorescent probe, RCya for Pd^2+^, was synthesized, as shown in Figure 1. Structures of the intermediates were characterized by IR, ^1^H NMR, and HRMS; their optical properties and the recognition mechanism of probe RCya were further confirmed by UV-Vis, HRMS, DFT, and fluorescence analyses.

Compounds **1–3** in Figure 1 were synthesized according to the reported methods [25]. The obtained compound **3** (0.0 29 g, 0.50 mmol) and 2,4-dihydroxybenzaldehyde (0.138 g, 1.00 mmol) were dissolved in methanol (25 mL), and the mixture was stirred at room temperature for 6 h. When completed, the reaction mixture was concentrated in vacuo and subsequently purified by column chromatography (silica gel, *V* (dichloromethane):*V* (ethanol) = 300:1 as eluent) to afford the probe RCya.

Probe RCya: 0.210 g, green solid; yield 48.26%. ^1^H NMR (400 MHz, CDCl_3_) δ: 1.11 (s, 6H, 2×-CH_3_), 1.43–1.46 (m, 6H, 2×-CH_3_), 2.32 (s, 3H, -CH_3_), 3.36 (t, J 3.5 Hz, 2H, -CH_2_-), 3.55–3.61 (m, 6H, 2×-CH_2_-, 2×-OH), 3.87–3.93 (m, 4H, 2×-CH_2_-), 6.50–6.53 (m, 4H, 3×=CH-, Ar-H), 6.62 (d, J 7.0 Hz, 2H, Ar-H), 6.76 (d, J 8.7 Hz, 1H, Ar-H), 7.02 (d, J 7.5 Hz, 2H, Ar-H), 7.11 (d, J 7.9 Hz, 1H, Ar-H), 7.23 (t, J 7.4 Hz, 1H, Ar-H), 7.42 (d, J 8.0 Hz, 1H, Ar-H), 7.86 (d, J 7.2 Hz, 2H, Ar-H), 8.16 (d, J 7.8 Hz, 1H, Ar-H), 8.53 (d, J 8.8 Hz, 1H, Ar-H), 9.27 (d, J 7.5 Hz, 1H, Ar-H). IR (KBr pellet, υ/cm^−1^): 3427 (υ_-OH_); 1742 (υ_C=O_); 1621 (υ_C=N_); 1516 (υ_C=C_); 1412 (σ_-OH_); 1326, 1214 (υ_C-O-C_). Elem. Anal.: calcd. for C_44_H_44_N_4_O_4_: C 76.28, H 6.40, N 8.09; found: C 76.43, H 6.60, N 8.15. HRMS (TOF, *m/z*): calcd. for C_44_H_44_N_4_O_4_ 692.3363; found [M+H^+^] = 693.3237.

### 4.3. Spectroscopic Analysis

Stock solutions (100 μmol L^−1^) of probe RCya, Pd^2+^, Sn^2+^, Cu^2+^, Ca^2+^, Mg^2+^, Cr^3+^, Cd^2+^, Co^2+^, Ba^2+^, Pb^2+^, Ag^+^, Fe^3+^, Fe^2+^, Ni^2+^, Al^3+^, Zn^2+^, Mn^2+^, Na^+^ and K^+^ were prepared in EtOH-H_2_O (*V* (EtOH):*V* (H_2_O) = 1:99, PBS, pH = 7.4). When used for spectroscopic tests, the stock solutions were usually diluted with EtOH-H_2_O (*V* (EtOH):*V* (H_2_O) = 1:99, PBS, pH = 7.4) to 10 μmol L^−1^ unless noted. All the measurements were performed at least in triplicate and averaged.

### 4.4. DFT Calculations

To evaluate the photo-induced luminescence process and verify the proposed mechanism of RCya–Pd^2+^ recognition, DFT calculations [26,27] were employed to construct the molecular model using 6–31G** for C/H elements, 6–31+G* for N/O/S elements, and DGDZVP for Pd on Gaussian 09 Program. The polarizable continuum model was applied to simulate the solution environment with ethanol as the solvent, and all the optimized structures were confirmed with no imaginary frequency.

### 4.5. Nanofiber Imaging

For the detection and imaging of Pd^2+^ by RCya on solid-state nanofibers, composite nanofibers that contained probe RCya and polyacrylonitrile (PAN M_w_ = 30,000, *m* (RCya):*m* (PAN) = 1:1000, DMF solution) as the matrix were prepared through the electrospinning technique. When used for nanofiber imaging, the RCya–PAN composite nanofibers were then immersed into a target ion-containing aqueous solution and imaged under ambient light or under the 365 nm ultraviolet lamp.

### 4.6. Cytotoxicity

Methyl Thiazolyl Tetrazolium (MTT) assay [28] was adopted to evaluate the toxicity of Pd^2+^ and RCya–Pd^2+^ in HeLa cells. HeLa cells in the exponential phase of growth were cultured in Dulbecco’s Modified Eagle Medium (DMEM) with 10% Fetal Bovine Serum (FBS). After digested with 0.25% trypsin solution, cells were dispensed in 96-well cell culture clusters (200 μL per well) at a density of 2.5 × 10^4^ cells per mL, incubated at 37 °C with 5% CO_2_ for 24 h. The probe RCya (100 mmol L^−1^ in DMSO) was then added to the 96-well plates and incubated for another 24 h to achieve a concentration gradient [29] of 6.25, 12.50, 25.00, 50.00, and 100.00 μmol L^−1^. Subsequently, the medium was removed, the cells were washed with PBS three times and incubated with 5.0 mg mL^−1^ MTT solution (20 μL MTT and 180 μL medium) at 37 °C for 4 h. After that, the cells were washed with PBS three times (1 mL per well) and then dissolved in DMSO (150 μL per well). The optical intensity at 490 nm, conducted in triplicate, was recorded on a microplate spectrophotometer. Data were expressed as mean ± standard deviation (SD).

### 4.7. Fluorescent Imaging in Living Cells

HeLa cells were cultured in 10% FBS-containing DMEM at 37 °C in the humidified atmosphere with 5% CO_2_. After 2 h, the growth medium was removed, and the cells were firstly washed with DMEM and incubated with 10 μmol L^−1^ of probe RCya for 30 min at 37 °C, then washed three times with PBS and imaged [30,31]. After subsequently adding 10 μmol L^−1^ Pd^2+^, HeLa cells were finally incubated with RCya and Pd^2+^ for 30 min at 37 °C, washed three times with PBS, and imaged. In the co-staining experiments of RCya–Mito/Lyso, HeLa cells were costained with 1 μmol L^−1^ RCya, 1 μmol L^−1^ Pd^2+^, and 100 nmol L^−1^ Mito/Lyso Tracker for 30 min at 37 °C, and then washed three times with PBS and imaged.

## Data Availability

The data presented in this study are available in the article.

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
