# Peer review of "A PET Fluorescent Probe for Dynamic Pd2+ Tracking with Imaging Applications in the Nanofiber and Living Cells"

_molecules, 2023, doi:10.3390/molecules28073065_

Round 1
Reviewer 1 Report
Dear Author
The manuscript is an interesting representation of solid-state RCya-PAN composite nanofibers and in living HeLa cells is excellent. But please explain the benefit and novelty of your work to previous publications like:
Wen J, Lv Y, Xia P, Liu F, Xu Y, Li H, Chen SS, Sun S. A water-soluble near-infrared fluorescent probe for specific Pd2+ detection. Bioorganic & Medicinal Chemistry. 2018 Feb 15;26(4):931-7.
Gong S, Zhou E, Liu Y, Gui Z, Feng G. A Pd2+-free near-infrared fluorescent probe based on allyl ether isomerization for tracking CORM-3 with high contrast imaging in living systems. Analytical Chemistry. 2022 Jan 21;94(4):2042-7.
Ahmmed E, Sarkar D, Mondal A, Saha NC, Bhattacharyya S, Chattopadhyay P. A new metal-free benzorhodol-based photoluminophore selective for carbon monoxide detection applicable in both in vitro and in vivo bioimaging. Analytical Methods. 2022;14(33):3196-202.
Reviewer 2 Report
This manuscript by Cheng et al. reports a PET fluorescent Pd2+ probe. This is a comprehensive study, I recommend a minor revision.
1. I suggest authors to modify the title of this manuscript, the current title is confusing.
2. To catch the eyes of readers, a scheme describing the main content of this study is recommended to be added at the beginning of this manuscript.
3. Scale bar is missing in Figure 11.
4. More details on detection limit calculation should be provided.
5. For DFT calculation, some parameters of calculation method should be presented.
